# Assessing Physical Fitness of Athletes in a Confined Environment during Prolonged Self-Isolation: Potential Usefulness of the Test of Maximal Number of Burpees Performed in 3 Minutes

**DOI:** 10.3390/ijerph19105928

**Published:** 2022-05-13

**Authors:** Joshua Qi Jun Tai, Shu Fen Wong, Steve Kin Ming Chow, Darine Hui Wen Choo, Hui Cheng Choo, Sofyan Sahrom, Abdul Rashid Aziz

**Affiliations:** 1Sport Science and Sport Medicine, Singapore Sport Institute, Sport Singapore, Singapore 397630, Singapore; joshuatai@snoc.org.sg (J.Q.J.T.); wongshufennn@gmail.com (S.F.W.); chow_kin_ming@sport.gov.sg (S.K.M.C.); darinechoo@snoc.org.sg (D.H.W.C.); kester_choo@sport.gov.sg (H.C.C.); 2Sport Physiology, Sport Science, National Youth Sport Institute, Singapore 397778, Singapore; sofyan_sahrom@nysi.org.sg

**Keywords:** maximal exercise, 3-MBT, home-based testing, self-testing tool, COVID-19, quarantined, isolation

## Abstract

Due to a prolonged period of quarantine during the COVID-19 pandemic, it is essential to monitor the physical condition of athletes isolated at home with a fitness tool that measures various aspects of physical fitness, which does not require any special equipment, and can be performed within a small space. This study assessed the reliability and validity of the test of performing the maximal number of burpees in 3 min (or 3-MBT) to monitor strength, power, and aerobic endurance in trained athletes. For Part I (reliability of the 3-MBT), 20 (10 male, 10 female) national athletes from various sports performed the 3-MBT on two separate test sessions. Athletes performed as many burpees as possible within 3 min and the primary performance criteria was the number of burpees completed (where a higher number reflected a better performance). The 3-MBT displayed excellent relative reliability in the athletes, with an intraclass correlation coefficient (ICC) and coefficient of variation (CV %) of >0.92, and <3.0%, respectively. For Part II (validity of the 3-MBT), 40 (20 M, 20 F) athletes performed the 3-MBT, and the countermovement jump (CMJ), isometric mid-thigh pull (IMTP), seated medicine ball throw (SMBT), isometric bench (IBP) and maximal aerobic power (VO_2max_) tests on separate sessions. When data of male and female athletes were pooled, there were significant correlations between relative 3-MBT and relative CMJ height (*r* = 0.65, large; *p* < 0.001), relative IMPT peak force (*r* = 0.50, large; *p* < 0.001), relative IBP peak force (*r* = 0.36, moderate, *p* = 0.02), and relative VO_2max_ (*r* = 0.50, large; *p* < 0.001). In conclusion, the 3-MBT is a reliable and reasonably useful test and is a valid assessment of lower body power and strength, upper body strength and aerobic fitness in trained male and female athletes.

## 1. Introduction

During the peak of the COVID-19 pandemic, globally, many athletes were mandated to stay at home. Whilst isolated, they were still expected to continue with their sports training and maintain their physical fitness levels. In many countries such as Singapore, strict stay-home advice was issued with no one allowed to leave their homes for a four-month period. In such extreme and prolonged isolated situations, it was uncertain whether athletes were able to maintain their optimal physical functions and capacity relative to their sports when the compulsory isolation period order was subsequently lifted and when they returned to sport. Indeed, the scientific literature has shown that the lockdown has had a negative influence on the physical fitness of adolescents and trained athletes in terms of strength, power, and aerobic endurance, despite evidence that the athletes were engaging in home-based training programmes [1,2].

While there have been other simple tests that can be performed at home or within small spaces, these tests are equipment-dependent and are one-dimensional in nature as they do not measure other aspects of the athlete’s physical attributes. For example, the step-test which is used to assess one’s cardiovascular fitness is dependent on a stool or chair of a specific height or the vertical jump test which assesses lower body power, which still requires equipment such as a measurement device or mobile phone application [3,4,5]. The push-up test assesses the upper body endurance but does not measure other aspects of the athlete’s physical attributes such as strength and power. Even though many of the aforementioned tests are valid and can be easily performed within a small space setting; again, they each, however, measure only a single aspect of the individual’s overall physical attributes. Therefore, a primary focus of this study was to examine the possibility of assessing several aspects of an individual’s physical fitness attributes with a single test tool without the use of specialized equipment and that can be performed within a small space of a home environment.

The burpee exercise has previously been used as a training tool in many sports and in the military [6,7]. The exercise is simple to perform and does not require equipment. The exercise is a whole-body calisthenic movement that taxes the overall muscular and cardiovascular systems and involves the motor skills of balance, agility, and coordination; albeit there has been no experimental study that has confirmed these assertions [6,8]. It has also been shown that the burpee exercise is multi-joint and activates many of the body’s primary muscle groups such as the chest, shoulders, quadriceps, hamstrings, and core [8,9]. Recently, Podstawski et al. [10] published normative data on performance for a 3-min all-out maximal effort burpee test, aptly termed the 3-Minute Burpee Test (or 3-MBT). They provided a percentile ranking of low, average, and superior performances in the test based on 5971 men and 3862 women between 18 and 25 years of age in Europe (individuals were from Poland, Great Britain, Hungary, and Serbia). The same authors claimed that the performance in the 3-MBT, i.e., the number of completed burpees in 3 min, seemed to assess the overall “strength-endurance” attribute of the individual, although the authors did not provide any experimental evidence to support this view [11,12,13]. In another study, the same investigators also showed a good association between performance in the 3-MBT with distance thrown with a medicine ball [11]. Interestingly, a study conducted by Sakamaki [9] showed that the burpee exercise (with an additional push-up movement) test has shown to be a useful indicator of aerobic endurance in young individuals, between 10 to 29 years old. The individuals’ exercise and post-exercise heart rate (HR) in the burpee exercise test showed positive correlations with the exercise and post-exercise recovery HR in a step-test [9]. Hence, there is evidence that 3-MBT seems to be able to assess varying or different attributes of physical fitness in healthy individuals.

Therefore, in the present study, we explored the possibility of the 3-MBT as a self-monitoring tool for tracking the overall physical attributes (i.e., power, strength, and cardiovascular fitness), in athletes who must isolate for a prolonged period. In trying to achieve this, it is important to examine the reliability (Part I) and validity (Part II) of the 3-MBT in an athlete population. It is hypothesized that the 3-MBT is a reliable and valid measure of the athlete’s physical fitness attributes of strength, power, and aerobic endurance. If the 3-MBT is indeed reliable and valid, it will also be a useful and convenient self-monitoring tool that can assess fitness without using specialized equipment and can be performed in small spaces, even during non-COVID times.

## 2. Methods

### 2.1. Subjects

Twenty (10 male, 10 female) and forty (20 male, 20 female) participants from the country’s national-level athletes volunteered to participate in the reliability (Part I) and validity (Part II) of this research project, respectively. Their physical characteristics are depicted in Table 1. It should be noted that the 20 participants in Part I were also involved in Part II. The inclusion criteria of participants were: (i) 17 to 40 years old, (ii) currently in training with the national team or squad, (iii) did not have any musculoskeletal injuries in the past 3 months. All athletes were training between 5 to 12 sessions per week, including their own sport-specific training and other auxiliary types of training such as resistance, anaerobic and aerobic conditioning. Athletes were instructed to refrain from intense training, tobacco, alcohol, and caffeine the day before testing sessions. Both the athletes and their parents (for those under 21 years old) provided written informed consent for the study, which was approved by the institutional ethics review committee, PH-FULL-035. Because the outcome of the present study was meant to be applied to athletes from all types of sports, it was deliberately a priori to recruit athletes from various sports including water- and ice-based, as well as sports that predominantly involved the upper body such as canoeing. The athletes recruited were from continuous endurance sports (3 male, 5 female), intermittent endurance sports (10 male, 8 female), short-duration explosive sports (2 male, 2 female) and combat/martial art sports (5 male, 5 female). Athletes were briefed on all tests’ protocols and possible risks before data collection. All data collection was conducted in the Singapore Sport Institute Human Performance Laboratory and Gymnasium facilities.

### 2.2. Experimental Design

Part I (reliability) of the study aimed to determine the inter-day reliability of the 3-MBT, to ensure that the test is replicable and reproducible. A test-retest experimental design was adopted where participants, after a familiarization session (first visit), performed the 3-MBT twice, in two separate sessions (second visit: Test 1 and third visit: Test 2), three to seven days apart, conducted at the same time of the day. For Part II (validity), the aim was to evaluate if the 3-MBT was a valid measure of strength, power, and aerobic endurance. Here, the performance in the 3-MBT was correlated to the same individuals’ performance in the countermovement jump (CMJ) and isometric mid-thigh pull (IMTP) as assessment of the lower body power and strength, respectively; and with seated medicine ball throw (SMBT) and isometric bench press (IBP) as assessment of the upper body power and strength, respectively; and finally with maximal aerobic power (VO_2max_) as a marker of cardiovascular or aerobic fitness.

### 2.3. Procedures

This study was conducted during the COVID-19 pandemic. Participants completed all the tests without wearing face masks and social distancing rules were adhered to. Participants visited the laboratory for three sessions each for both Part I and Part II, where the first visit was a familiarization (to the 3-MBT and all other measures) and the second (Test 1) and third visits (Test 2) were actual testing sessions. Each of the three sessions was separated three to seven days apart. During the familiarization session, the individuals’ body stature and mass were measured (IND 452-B150 Mettler Toledo, Germany), followed by familiarization with the burpee exercise. Here, after completing a standardized warm-up, the burpee exercise was demonstrated to the athlete, and instructions were clearly provided on the proper technique and cues on how to perform it correctly (see Figure 1). Athletes then practiced the burpee exercise until the investigator was satisfied that the athlete was able to perform it correctly and consistently. Thereafter, the athlete practiced the 3-MBT with maximal effort for a duration of 60 s to complete his/her familiarization with the 3-MBT.

Athletes who participated in Part II but were not involved in Part I underwent the same familiarization session, which included the 3-MBT, and all other physical tests such as CMJ, IMTP, SMBT, IBP and VO_2max_ tests. This was followed by two actual test sessions on sessions two and three, separated three to seven days apart. In session two, participants performed the tests in the following order: 3-MBT, CMJ and IMTP. In session three, they performed the tests in the following order: SMBT, IBP and VO_2max_. We incorporated adequate rest periods between the different tests to ensure physical fatigue was not a confounding factor in the athlete’s performance for each test.

### 2.4. The 3-Minute Burpee Test (or 3-MBT)

Before performing the 3-MBT, a standardized warm-up consisting of 3 min of easy cycling on an air-braked cycle ergometer (Wattbike Trainer, Nottingham, UK) followed by alternating shoulder tap (five repetitions on each side), inchworm to a downward dog position (five repetitions) and alternating reverse lunge (five repetitions on each side). All these exercises were meant to prepare the primary muscle groups used during the burpee exercise, which were the chest, shoulders, quadriceps, hamstrings, and core. An HR monitor (RS400; Polar Electro Oy, Kempele, Finland) was then strapped to the chest to assess HR during the test. In the 3-MBT (Figure 1), the athlete was instructed to perform as many burpees as possible within three minutes. Each burpee with the correct technique was considered as a completion of a successful burpee. Burpees completed without the correct technique were not counted. In the push-up position, arms should be fully extended with no bend, the individual’s back should not be arched or rounded, and legs should be fully extended with no knee bend. Only a valid or completed repetition was counted and athletes were warned if the repetition was not performed correctly and was not counted. The athlete was counted down every 30 s during the 3 min test duration. The performance criterion is the total number of burpees completed within 3 min where a higher number of burpees completed reflects a superior physical performance. At the end of the 3-MBT, peak HR was immediately recorded. At the 60 s post-test mark, blood lactate (BLa) was measured via finger prick (Lactate Pro; Arkray, Kyoto, Japan), and at the 3 min post-test mark, the ratings of perceived exertion (RPE; Borg category ratio 1–10) scale was administered.

### 2.5. Countermovement Jump (CMJ) Test

The CMJ test has been shown to be a valid and reliable assessment of lower body power [14]. It was performed on dual force plates (FD4000, ForceDecks, VALD Performance, Queensland, Australia) with a sampling rate of 1000 Hz. The athlete was instructed to keep his/her arms akimbo to eliminate arm swing and maintain an upright back to reduce any angular displacement of the hip. Three practice trials at ~50%, ~80% and ~100% of their perceived maximum effort were allowed. The athlete performed three actual test trials with maximal effort, each separated by 30 s of passive rest. The commercially available software (ForceDecks, Queensland, Australia) was used to generate and analyze the test variables and used a 20 N offset from the measured body mass, that was measured before the jump, to define the initiation of the jump. The athlete remained as still as possible for at least one second prior to the commencement of the countermovement action. The end of the eccentric phase and the start of the concentric phase was defined as the greatest negative displacement (absolute) and where velocity is equal to zero, which corresponds to force returning to body mass. Take-off was defined as the time point at which the total vertical force fell below the threshold of more than 20 N below bodyweight. The jump height was calculated based on velocity at take-off, using the impulse–momentum relationship. The jump height (in cm) was used as the criterion measure of lower body power.

### 2.6. Isometric Mid-Thigh Pull (IMTP) Test

The IMTP test has been shown to be a valid and reliable test of maximal lower body strength and is highly correlated to sporting performances [15]. The IMTP was performed on the same dual force plates as that in the CMJ test and with a customized rack. The same software as that in the CMJ test was also used to generate and analyze the test variables. The athlete was instructed to adopt a posture that was similar to the start of the second pull of the clean, which involved a knee flexion angle of 125–135°, and a hip flexion angle of 140–150° stance. A handheld goniometer was used to confirm the correct knee and hip angles before locating the bar height for each participant. The athlete gripped onto the bar with elbows fully extended. Wrist lifting straps were used to attach his/her hands to the bar so that grip strength would not be a limiting factor. Upon the tester’s command, the athlete pulled the bar, by driving his/her feet onto the floor, ‘as hard and fast as possible’. The athlete performed three practice trials at ~50%, ~80% and ~100% of his/her perceived maximum effort and maintained the tension for 3 s each time. For the actual test, the athlete performed at least two actual test trials of his/her maximal effort with the tension maintained for 5 s, and each trial separated by 2 min of passive rest. The difference between the two trials’ absolute peak force should be less than 250 N, otherwise, additional trials were conducted until two readings were within 250 N [15]. The highest force generated was reported as the absolute peak force (in N) and this was used as the criterion measure of lower limb strength.

### 2.7. Seated Medicine Ball Throw (SMBT) Test

The SMBT test has been shown to be a valid and reliable assessment of upper body power [16]. The test involved the athlete sitting against a wall with the entire back firmly pressed flat against the wall and legs fully extended straight-out, shoulder-width apart. The athlete held on to a medicine ball (3.0 kg for females and 5.0 kg for males) with both hands at chest level, prior to throwing the medicine ball as far as possible, at a trajectory angle of ~45° upwards, using only the force of the arms with minimal lower limb movement. Prior to the actual test, three practice trials at ~50%, ~80% and ~100% of his/her perceived maximum effort were allowed. For the test trial, the athlete performed 3 trials with maximal effort, each separated by 2 min of passive rest. The distance where the ball landed to the wall (measured in centimeters, cm) was recorded as the criterion measure for upper body power.

### 2.8. Isometric Bench Press (IBP) Test

This IBP test has been shown to be a valid and reliable assessment of the upper body strength [17]. The test was conducted using the same dual force plate system as that in the CMJ test. The same software as that in the CMJ test was also used to generate and analyze the test variables. The bench was placed on a force platform to collect the ground reaction force. The athlete laid supine on the bench, with his/her head, upper back, and buttocks in contact with the bench, and feet in contact with the weight plates used to prop up the bench so that the bench was horizontally leveled with the force plates, and hands positioned slightly wider than shoulder-width apart. The barbell was positioned above the chest level and positioned at a height that allowed the athlete to maintain a 120° elbow angle position. A handheld goniometer was used to confirm the correct elbow angle before locating the bar height for each athlete. Upon the tester’s command, the athlete was instructed to push the bar, by driving his/her shoulders and back into the bench, ‘as hard and fast as possible’. The athlete performed three practice trials at ~50%, ~80% and ~100% of his/her perceived maximum effort and with the tension maintained for 3 s each. For the actual test, the athlete performed at least two actual test trials of his/her maximal effort with the tension maintained for 5 s, and each trial separated by 2 min of passive rest. The difference between the two trials’ absolute peak force should be less than 250 N, otherwise, additional trials were conducted until two readings were within 250 N [15]. The highest force generated during the IBP test was reported as the absolute peak force (in N) and this was used as the criterion measure of upper body strength.

### 2.9. Maximal Aerobic Power (VO_2max_) Test

The VO_2max_ test was used as the criterion measure of aerobic fitness. Athletes who were in running-based sports such as basketball, squash and beach volleyball performed the test on the treadmill and those in non-weight bearing sports such as cycling, swimming, or sailing performed the test on a cycle ergometer. Athletes performed a standardized warm-up and self-stretching before the test commenced.

The running VO_2max_ test was conducted using an incremental run to exhaustion on a treadmill (Venus, HP-Cosmos, Germany). The test commenced with an initial speed of 10.0 km∙h^−1^ (for male) and 8.0 km∙h^−1^ (female) at a zero gradient for 60 s. After which, the speed was increased by 1.0 km∙h^−1^ every 60 s for the next 5 min; from here on, the treadmill gradient was increased by 1% every 60 s until a maximum of 8% elevation was attained. If the athlete was still able to continue, the treadmill speed was increased by 0.5 km∙h^−1^ every 60 s until the athlete attained volitional exhaustion. Volitional exhaustion was when the athlete signaled to terminate the run. Throughout the run, respiratory gas measures such as minute ventilation, oxygen uptake and expired carbon dioxide was measured with an open circuit spirometry system (TrueOne 2400 MMS, Parvomedics, East Sandy, UT, USA). HR was continuously monitored with a short-range telemetry monitor (Polar Electro OY, Kempele, Finland). After the completion of this initial VO_2max_ test, there was a 15 min passive rest before a verification phase was carried out to ensure that a “true” VO_2max_ value has been attained [18]. In this verification phase, the starting speed of the run was two stages before the termination stage of the initial test. The speed and gradient would then increase every 60 s in the same pattern as in the initial test until the athlete reached volitional exhaustion. The highest VO_2_ value obtained over any 20 s period in both the initial and verification tests was used in the analysis. The coefficient of variation for this test in our laboratory is 2% (unpublished data).

The cycle VO_2max_ was a progressive exercise ramp test on a cycle ergometer (Lode BV, Excalibur Sport, Groningen, The Netherlands). The athlete completed a standardized 3 min warm-up at 60 revolutions per minute (rpm). For the actual test, resistance started at 50 W and increased by 1 W every 3 s until volitional exhaustion was achieved. Volitional exhaustion was determined as the physical limit beyond which the athlete seized to exercise or was not able to maintain the prescribed pedal rate for a total of 5 s. Like the run test, there was also a 15 min passive rest before the verification phase. In the verification phase, the athlete cycled for 60 s each at 70%, 90% and 100% of the power output achieved in the initial test. Thereafter, he/she cycled at 120% of the terminated power output for as long as possible until volitional exhaustion. Respiratory gas measures and exercise HR were acquired as in the running test. The highest VO_2_ value obtained over any 20 s period was used in the analysis.

### 2.10. Statistical Analysis

All measures are expressed in mean (±SD). All statistical analyses were performed with SPSS v.24.0 for Windows (IBM SPSS Statistics, New York, NY, USA). The significance level for all statistical tests was set at *p* < 0.05. Prior to the analysis, all data sets were checked for their normality using histograms and Shapiro–Wilk tests. For Part 1 (reliability), paired sample *t*-tests were performed to compare the test’s measures (number of completed burpees, HR, BLa and RPE) between Test 1 and Test 2. Absolute reliability for performance in the 3-MBT between Test 1 and Test 2 was calculated using the Coefficient of Variation (CV) expressed as a percentage (%) and categorized as excellent (≤5%) and poor (>5%) [19]. Relative reliability was determined using the Intraclass Correlation Coefficient (ICC) and categorized as poor (<0.50), moderate (0.50 ≤ 0.75), good (0.75 ≤ 0.90) and excellent (>0.90) [20]. The ‘usefulness’ of the 3-MBT in its ability to detect changes in performance either over time or due to specific intervention was determined by comparing the Typical Error of Measurement (TEM) to the Smallest Worthwhile Change (SWC); and the interpretation was categorized as ‘good’ (when TEM < SWC), ‘ok’ (TEM = SWC) and ‘marginal’ (TEM > SWC) [21,22]. For Part II (validity), performance in the 3-MBT was correlated to performances in the various fitness tests using Pearson’s product-moment correlation coefficient (*r*). The interpretations of the strength of correlations were: small (0.1 ≤ |*r*| ≤ 0.29), moderate (0.3 ≤ |*r*| ≤ 0.49), large (0.5 ≤ |*r*| ≤ 0.69), very large (0.7 ≤ |*r*| ≤ 0.89), near-perfect (0.9 ≤ |*r*| ≤ 0.99) and perfect (|*r*| = 1.0) [23].

## 3. Results

For Part 1 (reliability), the descriptive statistics of the number of burpees completed and physiological responses during 3-MBT are shown in Table 2. The paired *t*-test indicated no significant differences between Test 1 and Test 2 in the number of completed burpees and the various physiological responses (i.e., HR, post-exercise BLa and RPE) in both male and female athletes (Table 2). The ICC, CV (%), TEM and SWC of the main performance measure of the 3-MBT, i.e., the number of completed burpees for male, female and pooled athletes are shown in Table 3. The absolute reliabilities across male, female, and pooled athletes were shown to be excellent (CV % = 2.6, 2.7 and 2.6, respectively). Similarly, the relative reliabilities across male, female, and pooled data were also excellent (ICC = 0.94, 0.93 and 0.94, respectively). The 3-MBT’s usefulness to detect change in males was deemed as ‘good’ (TEM 2.52 < SWC 2.57), and for females was ‘marginal’ (TEM 3.09 > SWC 1.85), and for the pooled data it was also deemed as ‘marginal’ (TEM 2.80 > SWC 2.31).

Table 4 showed the descriptive results of all the test measures taken in Part II (validity). Due to the high influence of body mass and body height on the energy costs of performing the burpees on the number of burpees completed, the results are reported in absolute and relative (adjusted to body mass index or BMI) terms [8,11,12]. Similarly, all the other physical fitness tests’ results of CMJ, IMTP, SMBT, IBP and VO_2max_ measures are also reported in absolute and relative terms (i.e., accounting for body mass or body height) [24,25,26]. Table 5 showed the paired correlations between the performance in the 3-MBT and the various fitness attributes. There were mixed results when assessing the bivariate correlations between performance in the 3-MBT and with the various fitness variable correlations, specifically analyzing the male and female athletes. However, when data of male and female athletes were pooled together, there were significant correlations between relative 3-MBT with relative CMJ height (*r* = 0.65, large; *p* < 0.001), and with relative IMPT peak force (*r* = 0.50, large; *p* < 0.001), and with relative IBP peak force (*r* = 0.36, moderate, *p* = 0.02), and with relative VO_2max_ (*r* = 0.50, large; *p* < 0.001). An unexcepted observation was the significant (moderate to large), but inverse correlations between relative performance in the 3-MBT and relative distance thrown in the SMBT.

## 4. Discussion

The purpose of this study was to determine the reliability and validity of the 3-MBT, with the aim of assessing the usefulness of the 3-MBT as a simple, i.e., can easily be performed within a very small space with no specialized equipment, self-testing, or monitoring tool of the athlete’s physical fitness attributes during prolonged periods of home self-isolation. The results showed excellent relative (ICC > 0.92) and absolute reliability (CV <3.0%) in the number of completed burpees. Part I (reliability) data thus indicated that the 3-MBT is a very reliable test. For Part II (validity), when data of male and female athletes were pooled, there were significant correlations between relative performance in the 3-MBT with relative CMJ height (*r* = 0.65, large; *p* < 0.001) and relative IMPT peak force (*r* = 0.50, large; *p* < 0.001), relative IBP peak force (*r* = 0.36, moderate, *p* = 0.02), and relative VO_2max_ (*r* = 0.50, large; *p* < 0.001). These results indicate that the 3-MBT is a valid measure of lower body strength and power, upper body strength and aerobic endurance in trained athletes. Taken together, these findings supported our hypothesis that the 3-MBT is a reliable and useful tool to assess the trained athlete’s lower body power, strength and aerobic fitness that can easily be conducted without the need for any sophisticated equipment and performed in a very small space during prolonged isolation at home.

The exercise HR, post-exercise BLa and RPE responses in the present 3-MBT indicate that the protocol involved high to maximal-intensity/effort exercise. These physiological responses are at levels equivalent to previous studies that have used the burpee movement modality but with differing protocols such as the 3 min all-out burpees [9]; three sets of 3 min all-out burpees [7]; and four sets of 30 s all-out maximal burpees [6]. Compared to the published international norms of performance in the 3-MBT, the number of completed burpees performed by the present study’s male and female athletes lies above the 95th percentile and would be categorized as “very good” [10]. This is not unexpected since our study’s participants were national-level athletes who are currently in training with the national team or squads, whilst that of Podstawski et al. [10] were non-athletic university students, who are likely to be of untrained status.

The ICC is commonly used to assess the reliability of a measurement or testing method, wherein values over 0.90 are regarded as excellent relative test–retest reliability. Tests with excellent ICCs exhibit good stability and consistency of measurement over time and low measurement error [20]. The findings shown in Table 2 and Table 3 indicate that the number of completed burpees in the 3-MBT demonstrated excellent reliability in both male and female athletes. This outcome agrees with those of the previous 3-MBT study in female university students (mean age 21 ± 0.7 y) [27]. However, inter-subject variability can potentially affect the result and overestimate ICC values in a typically heterogeneous population, and thus, the use of ICC alone is insufficient [19]. That is to say that measurements with excellent relative reliability do not necessarily ensure consistent intertrial performance. Calculations of the test’s CV (in %) are thus further recommended to obtain within-subject variation in addition to measuring ICCs and confirming the test’s absolute reliability [28]. In this regard, the low CV value of <3% clearly indicates that the present study’s 3-MBT possesses very reliable properties when performed by trained male and female athletes [19].

The athletes underwent only a short familiarization with the 3-MBT and there were no significant differences in the athletes’ performances between the two testing sessions (i.e., Test 1 and Test 2; Table 2), and there were excellent ICC (95% CI) and CV (%) values (Table 3). This indicates that the 3-MBT is a relatively simple test that can be implemented easily and quickly in an athletic population. It should be noted that our result is in sharp contrast with a previous study which suggested that five trials were required before individuals were able to obtain a reliable performance in the 3-MBT [27]. The stark differences could be due to the characteristics of the subjects used in the two studies. The subjects in Podstawski et al. [27] were female university students with unreported training status, and most likely to be untrained or possessing very low levels of fitness, i.e., strength, power, and cardio-endurance, and have had limited exposure to the maximal type of exercise regimen. In the present study, however, participants were national-level athletes with clearly superior levels of strength, power, and aerobic fitness. Furthermore, these athletes were highly likely to be more experienced in performing complex movement exercises and possessed higher physical and mental tolerance to maximal effort exercise. All of these could have assisted in ensuring very high reliability of the 3-MBT performance within their first test after an initial familiarization.

To assess the usefulness of the 3-MBT, i.e., to determine the practical relevance of the changes in the test because of any training or detraining intervention, it is critical to calculate the test’s SWC and TEM (see Table 3). In this regard, when combining the male and female data, the test’s TEM of 2.80 is slightly greater than the test’s SWC of 2.31, indicating that the 3-MBT is a ‘marginal’ test, i.e., it is not able to detect small changes but may be possible to detect a moderate level of changes in performance among trained athletes. It should be noted that the biological or technical variation or variability in performance in the 3-MBT is calculated as 2.8 burpees, or in real-world practice, 3 completed burpees. Thus, for the individual to have absolute confidence that a meaningful and true change in performance in the 3-MBT has occurred, there must be a difference of at least (2.8 × 2 = 5.6) 6 burpees between current and previous performance [28].

The burpee is a functional, weight-bearing exercise. The latter implies that those with a higher body mass, either fat or muscle mass, may be at a disadvantage when performing the burpees over time. This is supported by the observation that body composition and height have a negative relationship with the number of completed burpees performed in 3 min [11,12]. Thus, in the present study, we deliberately express the performance in the 3-MBT relative to athletes’ BMI (see Table 4). The burpee exercise movements involve flexion and extension of many joints and taxes the entire body musculature, especially the lower body or limbs. When squatting down to the ground (Stage B and C; Figure 1), there is clearly a large eccentric contraction on the lower limb muscles to absorb the force and subsequently, concentric contractions when pushing the body upwards to a standing position (Stage E and F, Figure 1). Consequently, the significant and large positive associations between relative performance in the 3-MBT and lower body power (i.e., relative CMJ height) and lower body strength (i.e., relative IMTP peak force) supports the view of a high involvement or contributions of the lower limb musculature in the 3-MBT performance. The data also showed a significant but moderately positive association between relative performance in the 3-MBT and upper body strength, i.e., relative IBP peak force. Indeed, during the burpee exercise, the arms are minimally involved with isometric contraction supporting the body’s position when on the ground (Stage C and D, Figure 1) and perhaps a slight push-off at the initial part to lift the body upwards (Stage E, Figure 1). Thus, with less involvement of the upper body (relative to the lower body), the modest correlation values between 3-MBT and upper body strength are thus not unexpected. The present study further showed a significant and large positive correlation of relative performance in the 3-MBT with relative VO_2max_, which suggests that aerobic fitness capacity is an important contributor to performance outcome in the 3-MBT. This view is supported by the close to maximal exercise HR and high levels of post-exercise blood lactate recorded during the 3-MBT (Table 2); where-in these values are concordant with values typically obtained during a VO_2max_ test. The high association between 3-MBT performance and VO_2max_ is further supported by the findings of Sakamaki [9], in which the author showed that the exercise and post-exercise recovery HR of the burpee exercise test correlated with corresponding exercise and recovery HR in an endurance-based step-test. Moreover, it has been shown that the 3-MBT had a strong positive association with enhanced levels of physical activeness among young men and women [29]. Thus, in summary, the overall findings of the excellent reliability and strong validity (significantly large correlations with lower body power and strength, modest correlation with upper body strength, and large correlation with aerobic capacity) indicate that the 3-MBT may be a useful single testing and monitoring tool to assess the various fitness attributes of an athletic population.

An unexpected finding of Part II of the study was the statistically significant inverse correlations between the distance thrown in the SMBT (as an index of upper body power) with performance in the 3-MBT (Table 5). This opposing relationship indicates that a weaker lower body power would result in a better performance in the 3-MBT, and vice-versa. The possible underlying reason or physiological mechanism(s) of this relationship is unclear, but it should be noted that the protocol of the burpee exercise used in the present study did not have a push-up component (see Figure 1), and this could have contributed to the resultant outcome. While in hindsight it may have been advisable to include the ‘push-up’ component in the present study, it was excluded for two reasons. The first was to be in sync with the previously published 3-MBT. The second reason was that during the piloting of this project, the ‘valid’ push-up movement, i.e., the depth that individuals would bend their elbows during the push-up movement, could not be consistently gauged with accuracy during the test.

There are several limitations of the present study that must be considered when interpreting the results of this study. The sample size of 20 male and 20 female athletes might be deemed as inadequate for Part II of this study in determining the validity of the 3-MBT with the many different fitness attributes. The assessment of a burpee completed with the correct technique may be deemed as somewhat subjective, but in the present study, the same researcher was involved in all the testing sessions of the 3-MBT. The present burpees exercise did not include a vertical jump component or a push-up component. The inclusion of both components may likely have a positive impact in increasing the contribution or involvement of the lower and upper body power and/or strength, respectively [30]. It should however be pointed out that the inclusion of both movements will further increase the subjective evaluation of a valid burpee. It is also important that further research is conducted to directly examine the sensitiveness of the 3-MBT to track and detect small changes because of specific training and/or dietary intervention.

## 5. Conclusions

The present results indicate that, among trained male and female athletes, the 3-MBT is very reliable, and only a difference of six or more burpees relative to previous performance would be deemed as a relevant and meaningful change in the subsequent assessment. The performance in the 3-MBT has moderate to strong associations with athletes’ lower body power and strength, upper body strength and aerobic fitness. Counterintuitively, upper body power is inversely related to performance in the 3-MBT. Nonetheless, there is clearly potential for the 3-MBT to be a single measuring tool that is simple to perform, requires hardly any equipment, can be performed in a very confined space, and with the ability to assess varying fitness attributes in athletes from various types of sports who are forced into quarantine or lock-down for a prolonged period.

## Figures and Tables

**Figure 1 ijerph-19-05928-f001:**
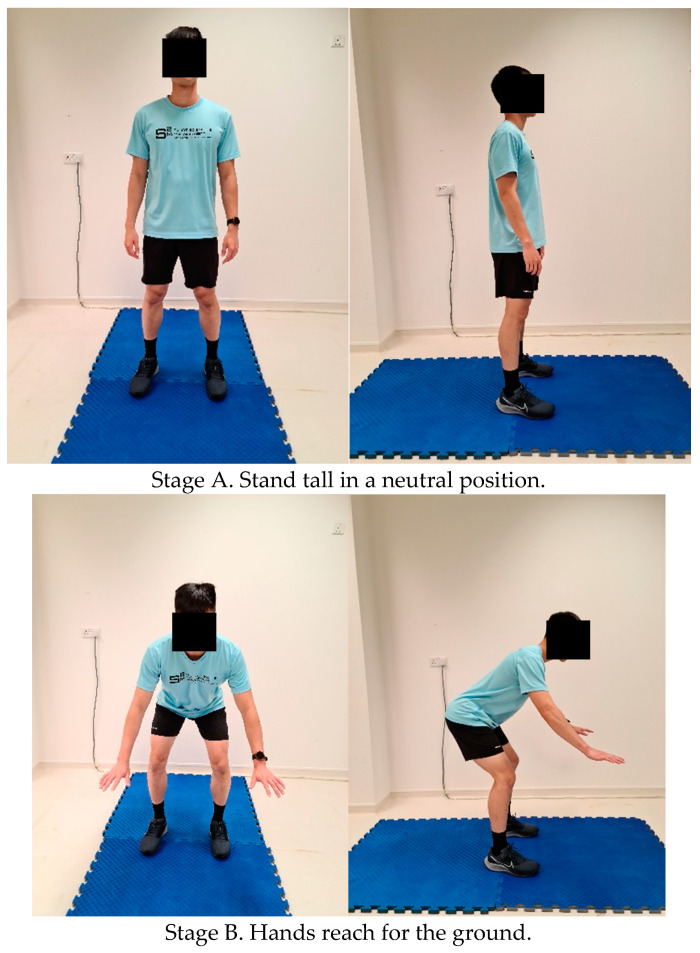
The study’s standard burpee movements with specific cues.

**Table 1 ijerph-19-05928-t001:** Physical characteristics of athletes involved in Part I (reliability) and Part II (validity) of the 3-min burpee test (3-MBT) study.

	Male	Female	Pooled
**Part I (reliability)**	(*n* = 10)	(*n* = 10)	(*n* = 20)
Age (y)	23.5 ± 4.2	21.9 ± 3.8	22.7 ± 4.0
Body mass (kg)	72.2 ± 8.5	59.4 ± 7.8	65.8 ± 10.3
Stature (cm)	173.7 ± 9.9	159.8 ± 6.0	166.7 ± 10.7
Body mass index (kg·m^−2^)	23.9 ± 1.8	23.2 ± 2.1	23.6 ± 1.9

**Part II (validity)**	(*n* = 20)	(*n* = 20)	(*n* = 40)
Age (y)	22.9 ± 4.2	23.0 ± 3.3	22.9 ± 3.7
Body mass (kg)	72.4 ± 9.6	58.8 ± 8.4	65.6 ± 11.3
Stature (cm)	176.3 ± 9.4	160.0 ± 6.8	168.1 ± 11.5
Body mass index (kg·m^−2^)	23.3 ± 2.2	22.9 ± 2.3	23.1 ± 2.2

**Table 2 ijerph-19-05928-t002:** Part I (reliability): Descriptive statistics of number of burpees completed and physiological responses during the 3-min burpee test (3-MBT) between Test 1 and Test 2 in male and female athletes.

	Male (*n* = 10)			Female (*n* = 10)			Pooled (*n* = 20)		
	Test 1	Test 2	*p*-Value	Test 1	Test 2	*p*-Value	Test 1	Test 2	*p*-Value
**Number of completed burpees**	87 ± 13	90 ± 11	*0.06*	80 ± 9	81 ± 13	*0.58*	84 ± 12	85 ± 13	*0.09*
**Heart rate at end of exercise (b·min^−1^) and as percentage of HR_max_ (%)**	170 ± 7 93.2 ± 5.6	169 ± 6 92.6 ± 4.5	*0.55* *0.52*	174 ± 8 94.6 ± 2.6	174 ± 9 94.5 ± 2.0	*0.90* *0.88*	172 ± 8 93.9 ± 4.4	171 ± 8 93.5 ± 3.5	*0.59* *0.56*
**Post-exercise blood lactate (mmol^−1^)**	12.2 ± 4.6	10.9 ± 4.8	*0.12*	10.9 ± 3.0	10.3 ± 3.7	*0.51*	11.5 ± 3.9	10.6 ± 4.2	*0.22*
**Post-exercise RPE (au)**	7 ± 2	7 ± 2	*1.00*	8 ± 1	8 ± 1	*0.59*	8 ± 2	8 ± 2	*0.68*

**Table 3 ijerph-19-05928-t003:** Part I (reliability): Coefficient of variation (CV), intraclass correlation coefficient (ICC) with 95% CI, technical error of measurement (TEM) and smallest worthwhile change (SWC) of the number of burpees completed in the 3-min burpee test (3-MBT) among male and female athletes.

	3-MBT: Number of Completed Burpees		
	Male (*n* = 10)	Female (*n* = 10)	Pooled (*n* = 20)
**Coefficient of variation (%)**	2.6(excellent)	2.7(excellent)	2.6(excellent)

**Intraclass correlation coefficient** **(with 95% CI)**	0.94(excellent) (0.75–0.99)	0.93 (excellent) (0.74–0.98)	0.94 (excellent) (0.85–0.98)

**Technical error of measurement**	2.52	3.09	2.80

**Smallest worthwhile change**	2.57	1.85	2.31

**Table 4 ijerph-19-05928-t004:** Part II (validity). Descriptive statistics of the number of burpees completed in the 3-min burpee test (3-MBT) and physical fitness assessments in absolute and relative (to body mass (BM) or to height) in male and female athletes.

	Male (*n* = 20)	Female (*n* = 20)	*p*-Value	Pooled (*n* = 40)
**Number of completed burpees** **Number of completed burpees (#·BMI^−1^)**	86 ± 15 3.7 ± 0.8	79 ± 12 3.5 ± 0.6	** *0.034 ** ** *0.17*	82 ± 14 3.6 ± 0.7

**CMJ height (cm)** **CMJ height (cm·kgBM^−1^)**	40.0 ± 4.2 0.6 ± 0.1	28.4 ± 3.5 0.5 ± 0.1	** *<0.001 ** ** * **0.025 *** *	34.2 ± 7.0 0.5 ± 0.1

**IMTP peak force (N)** **IMTP peak force (N·kgBM^−1^)**	2799 ± 412 38.9 ± 5.2	1865 ± 235 32.1 ± 4.2	** *<0.001 ** ** * **<0.001 *** *	2332 ± 577 35.5 ± 5.8

**SMBT distance (cm)** **SMBT distance (cm·height^−1^)**	408 ± 57 231 ± 24	334 ± 41 209 ± 21	** *<0.001 ** ** * **0.02 *** *	371 ± 62 220 ± 25

**IBP peak force (N)** **IBP peak force (N·kgBM^−1^)**	1745 ± 482 24.3 ± 6.8	1119 ± 235 19.2 ± 4.1	** *<0.001 ** ** * **0.02 *** *	1432 ± 491 21.7 ± 6.1

**VO_2max_ (L·min^−1^)** **VO_2max_ (ml·kgBM^−1^·min^−1^)**	4.03 ± 0.51 56.0 ± 6.0	2.52 ± 0.35 43.4 ± 7.5	** *<0.001 ** ** * **<0.001 *** *	3.27 ± 0.88 49.7 ± 9.2

# = number, BMI = body mass index, CMJ = countermovement jump, IMTP = isometric mid-thigh pull, SMBT = seated medicine ball throw, IBP = isometric bench press, VO_2max_ = maximal aerobic power, * *p*-value < 0.05 = significantly different between male and female athletes (highlighted in bold).

**Table 5 ijerph-19-05928-t005:** Part II (validity). Correlation analysis between the number of burpees completed in the 3-min burpee test (or 3-MBT) in absolute and relative (adjusted to BMI) value and physical fitness assessments in absolute and relative (to body mass (BM) or to height) in male (*n* = 20) and female (*n* = 20) athletes.

	Number of Completed Burpees			Number of Completed Burpees (#·BMI^−1^)		
	Male	Female	Pooled	Male	Female	Pooled
	Pearson’s *r* (strength of *r*) *p*-value	Pearson’s *r* (strength of *r*) *p*-value	Pearson’s *r* (strength of *r*) *p*-value	Pearson’s *r* (strength of *r*) *p*-value	Pearson’s *r* (strength of *r*) *p*-value	Pearson’s *r* (strength of *r*) *p*-value
**CMJ height (cm)**	−0.01 (small) *0.98*	0.02 (small) *0.95*	0.23 (small) *0.16*	0.15 (small) *0.54*	0.25 (small) *0.29*	0.27 (small) *0.10*
**CMJ height (cm·kgBM^−1^)**	**0.46** **(moderate)** ** *0.04* **	0.20 (small) *0.40*	**0.39** **(moderate)** ** *0.01 ** **	**0.69** **(large)** ** *<0.001 ** **	**0.59** **(large)** ** *0.006 ** **	**0.65** **(large)** ** *<0.001 ** **
**IMTP peak force (N)**	−0.29 (small) *0.22*	−0.09 (small) *0.72*	0.10 (small) *0.55*	−0.33 (moderate) *0.16*	−0.22 (small) *0.36*	0.06 (small) *0.10*
**IMTP peak force (N·kgBM^−1^)**	0.31 (moderate) *0.18*	0.27 (small) *0.24*	**0.39** **(moderate)** ** *0.01 ** **	0.44 (moderate) *0.053*	**0.57** **(large)** ** *0.009 ** **	**0.50** **(large)** ** *<0.001 ** **
**SMBT distance (cm)**	**−0.62** **(large)** ** *0.004 ** **	−0.35 (moderate) *0.14*	−0.24 (moderate) *0.15*	**−0.64** **(large)** ** *0.002 ** **	**−0.56** **(large)** ** *0.01 ** **	**−0.36** **(moderate)** ** *0.02 ** **
**SMBT distance (cm·height^−1^)**	**−0.52(large)** ** *0.02 ** **	−0.09(small) *0.70*	−0.18 (small) *0.28*	**−0.64(large)** ** *0.003 ** **	−0.35 (moderate) 0.13	−**0.37** **(moderate)** **0.02 ***
**IBP peak force (N)**	−0.17 (small) *0.47*	0.04 (small) *0.86*	0.09(small) *0.58*	−0.14 (small) *0.55*	−0.14 (small) *0.56*	0.03 (small) *0.88*
**IBP peak force (N·kgBM^−1^)**	0.15 (small) *0.52*	0.31(moderate) *0.19*	0.29 (small) *0.07*	0.26 (small) *0.27*	0.44 (moderate) *0.054*	**0.36(moderate)** ** *0.02 ** **
**VO_2max_ (L·min^−1^)**	−0.41(moderate) *0.07*	0.22 (small) *0.36*	0.14 (small) *0.38*	−0.43 (moderate) *0.06*	0.05 (small) 0.83	0.04 (small) *0.79*
**VO_2max_ (ml·kgBM^−1^·min^−1^)**	0.22 (small) *0.36*	**0.48** **(moderate)** ** *0.03 ** **	**0.42** **(moderate)** ** *0.007 ** **	0.41 (moderate) *0.08*	**0.68(large)** ** *<0.001 ** **	**0.50** **(large)** ** *<0.001 ** **

# = number, BMI = body mass index, CMJ = countermovement jump, IMTP = isometric mid-thigh pull, SMBT = seated medicine ball throw, IBP = isometric bench press, VO_2max_ = maximal aerobic power, * *p*-value < 0.05 = significantly correlated between performance in the 3-MBT and fitness performance measures (highlighted in bold).

## Data Availability

Data of the study may be requested from the corresponding author.

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
