# Peer review of "Assessing Physical Fitness of Athletes in a Confined Environment during Prolonged Self-Isolation: Potential Usefulness of the Test of Maximal Number of Burpees Performed in 3 Minutes"

_ijerph, 2022, doi:10.3390/ijerph19105928_

Round 1

Reviewer 1 Report

Review: Assessing various attributes of physical fitness of athletes in a 2 confined environment during prolonged self-isolation: Potential usefulness of the test of maximal number of burpees performed in 3 minutes

This study examines burpees and the potential for them to be a single and simple measuring tool to assess fitness.

(Page/Line)

1/28: Remover “very”

2/81: Remove “Japanese Author”

3/104: Replace “five” with 5 since 12 is in the same sentence

Table 1: Cleaned up. Gaps in one area and not in another

6/179: Remove period

Tables 4 and 5 are confusing. Cleaning them up would clear up confusion. For example, the pooled data seems to be clearest so why not have that in the table and explain the other data briefly. Tables do not clearly and cleanly aid in understanding the data.  

Reviewer 2 Report

Thank you for the opportunity to review this study outlining the usefulness of the 3-BMT to assess fitness during isolation. I have the following edits and recommendations;

  1. Throughout the manuscript 'burpees' (plural) is used instead of 'burpee' (singular); just go through and double check where plural and where singular are appropriate. It seems it is a burpee test ? versus a burpees test?
  2. Suggest making the results applicable to any scenario where self assessment would be appropriate versus just a COVID situation; this will expand usefulness of results.
  3. Line 11 - change to 'quarantine' (remove the 'd')
  4. Line 15 - remove 'various physical fitness attributes' as seems redundant here
  5. Line 26 - remove 'the following fitness measures:' 
  6. Line 35 change 'all over the globe' to 'globally'
  7. Line 53 remove 'Another example is' and just state 'The push-up test......'
  8. Line 56 remove 'above mentioned' and use 'aforementioned'
  9. Line 64 remove 'any'
  10. Line 65 remove 'taxed on the individual's' and replace with 'tax'
  11. Lines 89-90 remove 'self-help tool in tracking and monitoring' and change to 'self-monitoring tool in tracking the overall physical......'
  12. Line 91 remove 'are being forced into isolation' and change to ' must isolate'
  13. Line 139 remove the 's' from 'techniques' so it reads 'technique;' also, what were the technique cues (the statements in the figure captions)? Technique is very important and is the #1 criticism of why most strength and conditioning specialists do not use a burpee anymore (most people cannot do them correctly for very long)
  14. Line 143 Clarify if familiarization was done on the same day from testing, or how far apart sessions were
  15. Did your participants complete the tests with a mask when in the lab? (The photo shows, but I'm not sure if this was for the sake of the photo session safety or if participants did this or not when in the lab)
  16. Lines +/- 170 What was the order of the tests?
  17. Lines 173 +  Was proper technique any part of the test criteria (burpee not counted because it was not done correctly?)?
  18. For all tests measuring force, identify if dual or one plate; and how if using one plate the bench stayed level (I'm imagining it was on an incline in order for the head of the bench to rest on a single plate); also, you cite ForceDecks as the company for the plates in Line 194 - that's a software company; did you use Vald uniaxial plates with the ForceDecks software?
  19. Line 301 Paired samples t-test would be if same participants performed the tests; maybe I am reading incorrectly, but it seems you had 2 different groups perform the tests. Independent t-tests would be appropriate then. Can you justify / clarify?
  20. Lines 315 and 316 The correlation ranges should be expressed as absolute values |                  | since values could be negative
  21. Tables - seems 'nos' is a little confusing; I see you stated it means number, but maybe just using the '#' symbol then stating # = number?
  22. In the Discussion address any limitations of the study (was technique not part of the criteria for assessing # burpees, for example).

Reviewer 3 Report

I have read with interest this paper where the Authors report data regarding the usefulness of the test of maximal number of burpees to monitoring the overall physical attributes in athletes forced to long isolation period, in pandemic time.

I would have a request to take into consideration to shorten the title that is really too long and a little redundant.
